# Immunogenicity, Efficacy, and Effectiveness of Two-Dose and Shorter Schedules of Hepatitis E Vaccine: A Systematic Review

**DOI:** 10.3390/vaccines13010028

**Published:** 2024-12-31

**Authors:** Bilal Azam, Melanie Marti, Amit Goel, Rakesh Aggarwal

**Affiliations:** 1Department of Gastroenterology, Jawaharlal Institute of Postgraduate Medical Education and Research, Puducherry 605006, India; bilalazampgi@gmail.com; 2Immunization, Vaccines and Biologicals, World Health Organization, 1211 Geneva, Switzerland; 3Department of Hepatology, Sanjay Gandhi Postgraduate Institute of Medical Sciences, Lucknow 226014, India; agoel.ag@gmail.com

**Keywords:** hepatitis E, HEV 239, shorter schedule, immunogenicity, efficacy

## Abstract

**Background**: Hepatitis E virus (HEV) is a leading cause of acute viral hepatitis in adults. The schedule for HEV 239, the only approved anti-HEV vaccine, consists of three doses at 0, 1, and 6 months, which is unsuitable for use in emergency and outbreak situations where quick protection is desired. We, therefore, undertook a systematic review of data on immunogenicity, efficacy, and effectiveness of alternative accelerated schedules. **Methods**: Data sources on immunogenicity, efficacy, and effectiveness of the HEV 239 vaccine following accelerated schedules published between 22 January 2005 and February 2024 were identified from five electronic databases, and the relevant data were extracted. **Results**: The search identified seven relevant reports, including one phase II pre-licensure trial, three reports from the phase III licensure trial, and three post-licensure reports. In these studies, following administration of the HEV 239 vaccine in two doses at 0 and 1 month or a three-dose rapid (0, 7, and 21 days) schedule, anti-HEV antibody seroconversion rates were similar to and geometric mean concentrations of anti-HEV antibody were only slightly lower than those following the standard three-dose schedule. In individuals who were seropositive for anti-HEV antibodies at baseline, the antibody response persisted for several years irrespective of the number of vaccine doses, and in those who were seronegative at baseline, administration of two vaccine doses induced antibodies whose level remained substantially high till at least 13 months of follow-up. Administration of two doses was also associated with a high protective efficacy against HEV infection and associated disease. **Conclusions**: The available data indicate that two doses of HEV 239 administered one month apart confer sufficiently high antibody titers and protection for at least 13 months, a duration which should be adequate for its use as an outbreak control measure.

## 1. Introduction

Hepatitis E is the most common cause of acute viral hepatitis in adults. It is caused by infection with hepatitis E virus (HEV), which has 27–34 nm diameter virions with a single-stranded RNA [1]. The viral genome has three open reading frames [1]; of these, the ORF1 encodes for the viral non-structural proteins, ORF2 for the capsid protein, and ORF3 for a protein with multiple functions. The virus has eight genotypes, four of which (namely, genotypes 1–4) are most frequently associated with human infections [2,3]. Genotypes 1 and 2 HEV affect only humans; by contrast, genotypes 3 and 4 HEV primarily infect pigs and several other mammalian species and are only occasionally transmitted to humans [2].

Clinical consequences of HEV infection vary from asymptomatic infection to typical acute hepatitis and occasionally acute liver failure, especially in pregnant women [4]. An infrequent presentation is chronic HEV infection, observed with genotypes 3 and 4 viruses, especially among those with immunosuppression [5].

HEV 239 (Hecolin^®^) is a hepatitis E vaccine that contains a recombinant 239-amino acid protein corresponding to amino acid 368–606 of ORF2 protein of genotype 1 HEV, expressed in *Escherichia coli* [6,7]. It is approved in China (2011) and Pakistan (2021) and has recently received marketing authorization in India. The recommended schedule is three intramuscular doses of 0.5 mL each (containing 30 µg of HEV antigen) at 0, 1, and 6 months among persons aged 16 years or above. This regimen is known to induce a strong antibody response and provides good protection against hepatitis E for at least 10 years [8]. The use of this three-dose (0, 1, 6 months) schedule was based on the results of a phase II study [9], in which this schedule induced a higher antibody titer compared to a two-dose schedule (0, 6 months), though the seroconversion rates with the two schedules were similar [9].

An important use case for this vaccine is for the control of HEV outbreaks [10], as exemplified by its recent use in an internally displaced population in Bentiu, South Sudan [10]. However, the three-dose schedule spread over 6 months is difficult to implement during outbreaks, especially when these occur in areas with mobile populations, conflict situations, and humanitarian emergencies. For instance, in the South Sudan campaign, though 86% of the eligible population took the first dose, the coverage for the second and third doses was only 73% and 58%, respectively [10].

Thus, there is a need for alternative schedules of this vaccine with fewer doses or a more rapid time-course. The regimens with fewer doses, besides being more practical and easier to administer, will also be less costly. Further, the shorter schedules should also be useful for emergency responders and international humanitarian workers, who need to be rapidly deployed.

We, therefore, decided to undertake a systematic review of the available data on immunogenicity, efficacy/effectiveness, safety, and duration of protection following the HEV 239 vaccine when administered in shorter-duration schedules with fewer than three doses or with reduced time intervals between doses.

## 2. Materials and Methods

Relevant articles were identified through a search of several databases [Medline OvidSP, EMBASE, Cochrane CENTRAL, PubMed, and International Clinical Trials Registry], using the following search terms: ‘Hepatitis E vaccin*’; ‘HEV 239’; ‘Hecolin’; ‘Hepatitis E/prevention and control’; and ‘Hepatitis E*/immunolog*’ on 22 February 2024. Titles and abstracts of the articles so identified were screened using the inclusion and exclusion criteria in Table 1. From the selected studies, data on seroconversion rates and antibody titers at different time points for various schedules were extracted by two reviewers.

## 3. Results

### 3.1. A Literature Search

The initial search identified a total of 47 records (Figure 1). After two reviewers (BA, RA) screened their titles and abstracts, 14 sources were identified for full-text assessment. Of these, seven papers met the inclusion criteria and were further analyzed; these are summarized in Table 2. A list of data sources that were excluded is available as Appendix A.

### 3.2. Immunogenicity

The data on antibody responses following one or two doses or following rapid three-dose regimens of the HEV 239 vaccine in humans were available from reports on pre-licensure phase II [9] and phase III [6,11,12] trials during vaccine development and from three reports on post-licensure studies [13,14,15]. Some of these reports were studies using the standard three-dose schedule that also provided data at a timepoint before the third vaccine dose had been administered or on study participants who, for some reason, did not receive all three planned doses.

#### 3.2.1. Pre-Licensure Phase II Trial

This study included healthy persons aged 16–65 years (33% women) in China who were negative for anti-HEV antibodies [9]. It had two parts: a dose-escalation component to determine the optimum amount of antigen in each dose and a dose-scheduling component to determine the optimum number of vaccine doses.

In the dose-scheduling study, 457 persons (mean age: 32.8 s ± 12.5 years) were randomly assigned to receive two doses (20-μg each, at 0 and 6 months) of HEV 239, three doses (20-μg each, at 0, 1 and 6 months) of HEV 239, or three doses of hepatitis B vaccine (5-μg each, at 0, 1 and 6 months; control group) by intramuscular route. Immunogenicity was assessed using seroconversion rates and geometric mean concentrations (GMC) of anti-HEV antibody, using the Wantai anti-HEV assay at months 0 (baseline), 2 (one month after the second dose in the three-dose group), 6 (before the final dose), 7 (one month after the final dose), and 13 (7 months after the final dose).

The seroconversion rates at 7 months with the two-dose and the three-dose schedules were similar (98% vs. 100%) (Table 3). However, the former schedule was associated with significantly lower anti-HEV antibody titers [GMC (95% CI) = 8.6 (6.5–11.3) WU/mL] than the three-dose schedule [15.9 (13.8–18.2); *p* < 0.05], with a GMC ratio (after two doses versus after three doses) of 0.54 (Table 3).

#### 3.2.2. Phase III (Licensure) Trial

In this large field trial in China [6,11,12], 112,604 healthy adults of either sex aged 16–65 years (mean [standard deviation] age = 44.7 [11.1] years; 56.5% female), irrespective of their HEV serostatus (47% seropositive at baseline), were randomized to receive three intramuscular doses (at 0, 1, and 6 months) of either HEV239 (30-µg each) or a hepatitis B vaccine (5-µg each), and were followed up for safety and efficacy (occurrence of clinical hepatitis E) beginning one month after administration of the third dose. In addition, in a subset of 11,165 subjects (‘the immunogenicity subset’, including 5567 in the HEV 239 arm and 5598 in the comparator HBV vaccine arm), blood specimens were collected at various time points up to 55 months after the first dose and tested for anti-HEV antibody titers using the Wantai quantitative assay.

Of the 5567 persons in the HEV 239 arm in the immunogenicity subset, some failed to complete the intended three-dose schedule but were, nevertheless, followed up as originally planned. Thus, 184 (3.3%) subjects missed the third dose (receiving two doses at 0 and 1 months, respectively), and 189 (3.4%) missed the second dose (receiving the first and the third doses at 0 and 6 months, respectively). For these participants, antibody titer data at the 43-month follow-up (i.e., 3 years after the completion of one month since the last planned dose) were available for 117 and 125 participants, respectively (data at 55 months were available for very few participants and, hence, were not reported) (Table 4 and Figure 2).

In individuals with baseline anti-HEV seropositivity (Figure 2a), GMC for anti-HEV remained high at 43 months of follow-up, irrespective of whether they had received one, two, or three doses of the vaccine.

By contrast, in individuals who were seronegative for anti-HEV at baseline (Figure 2b), the peak anti-HEV antibody concentration at 7 months (i.e., one month after completion of the planned schedule) and 3 years thereafter (i.e., 43 months from the start of vaccination) were somewhat lower in those who received two doses (whether 0–1 month, or 0–6 month) than those following the three-dose schedule, but were higher than those who had received only one dose or the placebo.

#### 3.2.3. Phase I Study in the United States

In a phase I study performed in the United States [15] in 2019, healthy volunteers (*n* = 25; age: 18–45 years; 19 [76%] women, all seronegative for anti-HEV antibody at baseline) were randomized to receive either HEV 239 vaccine (*n* = 20) or a placebo (*n* = 5) by intramuscular route at 0, 1, and 6 months. They were followed up to 6 months after the third dose. The seropositivity rate one month after the second dose was 100%, and GMC was 6.16 (1.68–18.77) WHO Units (WU)/mL. These anti-HEV levels were only somewhat lower than those achieved 1 month after the third dose [11.5 (2.72–33.02)] WU/mL, with a ratio of GMCs after two doses and after three doses being 0.54.

#### 3.2.4. Two-Dose (0–1 Month) Schedule Study

In a randomized, controlled trial in Bangladesh, designed to specifically study a two-dose HEV 239 schedule [14], 100 individuals (mean age = 26.1 ± 9.6 years; 52% males; 33% seropositive for anti-HEV antibody at baseline) were randomized to receive two intramuscular doses (at 0 and 1 month) either of HEV 239 or hepatitis B vaccine and were followed up for 24 months from the first dose. The seropositivity rates following the two-dose schedule of HEV 239 at 2 and 24 months from the first dose were 100% and 98%, respectively, and the GMC for anti-HEV antibody was 49.5 (31.5–77.7) WU/mL and 6.4 (3.6–11.3) WU/mL, respectively (Table 5).

#### 3.2.5. Rapid (0–7–21 Day) Three-Dose Accelerated Schedule Study

This randomized trial in China assessed the immunogenicity of an accelerated three-dose HEV 239 schedule administered over three weeks [13]. In this trial, 126 healthy individuals (mean age = 49.83 ± 10.67 years; 48% male), seronegative for anti-HEV, were randomized to receive three doses of HEV 239, in either an accelerated schedule (0, 7, and 21 days; *n* = 63) or the standard schedule (0, 1, and 6 months; *n* = 63). At seven months after the first dose, seropositivity rates in both groups were 100%, whereas GMC was somewhat lower with the accelerated schedule [5.0 (4.5–6.5) WU/mL] than the routine schedule [9.5 (8.0–10.1) WU/mL], with a GMC ratio of 0.53. The GMC antibody titer after the second dose of the routine schedule [5.0 (4.0–6.0) WU/mL] was similar to that after the rapid three-dose schedule.

### 3.3. Vaccine Efficacy

Data on vaccine efficacy were available from the pre-licensure phase II [9] and phase III [6,11,12] trials and from a post-licensure study in Bangladesh [14].

#### 3.3.1. Pre-Licensure Phase II Trial

This trial, though primarily meant to assess vaccine immunogenicity, also assessed the rates of new HEV infections [9]. For this, the HEV-seronegative study participants who had received three-dose schedule of HEV 239 vaccine (0, 1, and 6 months; *n* = 155), two-dose schedule of HEV 239 (0 and 6 months; *n* = 151), or three-dose schedule of the control hepatitis B vaccine (0, 1, and 6 months; *n* = 151) were followed up for evidence of HEV infection for 6 months after the last dose. HEV infection was defined as the occurrence of a greater than three-fold rise in the levels of IgG anti-HEV antibody in paired sera collected at 7 and 13 months, thereby excluding any antibody rise that occurred during the first seven months and could be related to the vaccine itself.

In all, nine instances of HEV infection were recorded in the control group and one each in the two-dose and the three-dose HEV vaccine groups. All these infections were sub-clinical. Thus, the protective efficacy against HEV infection following the two-dose schedule (85%; 95% CI = 10–99%) was comparable to that with the three-dose schedule (89%; 31–100%).

#### 3.3.2. Phase III (Licensure) Trial

This trial is the only study that had an assessment of the clinical efficacy of HEV 239 as the primary objective [6,11,12]. The primary efficacy analysis (per protocol) compared the rate of hepatitis E illness during the period from one month after the third dose till the end of follow-up in the participants who had received three doses of vaccine or placebo. Hepatitis E illness was defined by (i) occurrence of constitutional symptoms lasting at least three days, (ii) serum ALT elevation (at least 2.5-fold the upper limit of normal), and (iii) evidence of recent HEV infection (anti-HEV IgM or HEV RNA; and a ≥four-time increase in anti-HEV IgG concentration) [11].

In the most recent report, the trial participants in the per-protocol cohorts were followed up for 113 months beginning one month after the 6-month dose to obtain data on long-term efficacy [6]. In this report, in addition to the participants in the original study, residents in the 11 townships covered by the surveillance system who had not been enrolled in the trial and had been born between 1941 and 1991 (similar to the participant group) were included as an additional external control cohort (*n* = 178,236) to further evaluate vaccine efficacy.

In the per-protocol analysis, the protective efficacy of HEV 239 against HEV-associated illness was 89% (76–96%) at 102 months and 87% (73–94%) at 120 months of follow-up. Further, in a modified intention-to-treat analysis (those participants who had received any dose of HEV 239 vaccine and were followed up for efficacy up to 120 months from the beginning of vaccination), too, the efficacy was high during the extended follow-up, i.e., 85% (72–93%) at 102 months and 83% (69–91%) at 120 months.

An additional analysis evaluated vaccine efficacy after two doses (at 0 and 1 months) of the vaccine by combining data (i) for the participants who had received only two doses (followed up from 15 days after the second dose up to 120 months) for the entire follow-up period and (ii) for the participants who received three doses during the follow-up period beginning 15 days after the second dose till administration of the third dose. The participants (*n* = 250) who had received only the first and the third doses (i.e., two doses at 0 and 6 months) were excluded from this analysis. In this analysis, six cases of acute hepatitis E were recorded among placebo recipients (*n* = 3765 between 6 months to 30 months, and *n* = 52,428 till administration of the third dose) and none among the HEV 239 vaccine recipients (*n* = 3542 between 6 months to 30 months, and *n* = 52,235 till administration of third dose), indicating efficacy of 100.0%, albeit with broad 95% confidence intervals (15.4–100.0%) [6].

During the follow-up from 30–120 months after the start of vaccination, the number of new hepatitis E cases in the placebo group was too few, leading to inadequate statistical power for the estimation of the long-term efficacy of the two-dose regimen. However, a comparison of the number of cases among two-dose vaccine recipients with those in the post-hoc external control cohort showed a high efficacy (standardized for age and sex) rate (Table 6).

Of the 13 cases of hepatitis E that occurred in the placebo group in this trial in whom virus genotype could be identified, 12 had infection with genotype 4 HEV and one with genotype 1 HEV; hence, the evidence for the protection afforded by the vaccine was primarily for genotype 4 HEV [11].

#### 3.3.3. Two-Dose Schedule Study

In this Bangladesh trial, 100 individuals were randomized into two groups [HEV 239 vs. HBV control] (*n* = 50 each) and followed up to 23 months after the last dose for episodes of subclinical HEV infection. Five cases of HEV infection were observed in the 45 controls and none in the 43 subjects who received HEV 239, with a protective efficacy against HEV infection of 100% (82–100%).

### 3.4. Vaccine Effectiveness

No data are available on the effectiveness of HEV 239 in real-life settings, whether for a three or two-dose schedule.

### 3.5. Vaccine Safety

Data on the safety of a two-dose schedule of HEV 239 are available only from a pre-licensure study [9] in China with a head-to-head comparison of two-dose (0, 6 months) vs. threes-dose (0, 1, 6 months) schedules (each dose of 20 µg instead of 30 µg in the marketed product) [9]. In this study, there was no significant difference in the frequencies of all local (5.2 and 8.5 per 100 vaccine doses, respectively) or systemic (4.9 and 7.6 per 100 doses, respectively) adverse events or severe (grade 3 or above) local (0 and 1.6 per 100 doses) or systemic (0 in both) adverse events.

In another study [13], where an accelerated three-dose schedule (0, 7, and 21 days) was compared with the routine three-dose schedule (0, 1, and 6 months) of HEV 239 vaccine, the rates of solicited local (8/62 [12.9%] vs. 8/63 [12.7%]) and systemic (12/62 [19.4%] vs. 11/63 [17.5%]) adverse events were similar. The rates of unsolicited adverse events were 16/62 (25.8%) and 10/63 (15.9%), respectively, with no significant difference. No serious adverse event was observed in either group.

## 4. Discussion

Our systematic review of immunogenicity data following intramuscular administration of two doses of HEV 239 vaccine showed a high seroconversion rate similar to that following the three-dose schedule and anti-HEV antibody titers, which were somewhat lower than those following the conventionally recommended three-dose schedule. Similar results were also seen following administration of three doses in a rapid 21-day schedule. Further, two vaccine doses were associated with a good protective efficacy rate for at least one year.

There is sufficient evidence to indicate that anti-HEV antibodies are protective against HEV infection [16]. However, the titer defining the protection threshold remains unknown. Despite the lack of data on the protection threshold for antibody concentration, we believe that reasonably robust conclusions are possible from our analysis. Further, the trial data primarily support protection against genotype 4 HEV, a heterologous strain of the genotype 1 protein contained in the vaccine [11]. However, there are in vitro data that anti-HEV antibodies provide cross-protection across HEV genotypes [17,18].

In the pre-licensure phase II study [9], a two-dose schedule showed a seroconversion rate similar to that with the three-dose schedule (98% vs. 100%), albeit with slightly lower anti-HEV antibody titers (GMC = 8.6 vs. 15.9 WU/mL). However, the inter-dose interval in the two-dose schedule in this study was 6 months, and the schedule was no shorter than the conventional three-dose schedule. Further, albeit less importantly, each dose contained a smaller amount (20 µg) of HEV antigen than the marketed preparation (30 µg); it is possible that the results with the latter dose could have been different. These considerations limit the value of these data. However, these data are of importance because these represent the only randomized comparison of two-dose and three-dose schedules. Importantly, in this comparison, the GMC ratio of the titers in the two groups was 0.54. In this context, it is important to consider that in trials comparing different schedules for various vaccines, GMC ratios of above 0.50 are often considered non-inferior [16].

The most relevant data on the value of a shorter-dose schedule are those from the phase III licensure field trial, which have been published as three reports with 19-month, 5-year, and 10-year follow-ups, respectively, from the start of vaccination [6,11,12]. In this trial, some participants did not receive the intended three doses, creating two distinct subsets of participants who had received only two doses, i.e., (i) the planned first and second doses (at 0 and 1 month) or (ii) the planned first and third doses (at 0 and 6 months), which could be compared with those who had received all three planned doses. Of these, the former subset (0, 1 month), being a shorter-duration schedule, is of primary relevance. Though such post-hoc comparisons being non-randomized have limitations, they provide useful information about the response expected following the administration of two doses of the HEV vaccine.

In this trial, in individuals with baseline anti-HEV seropositivity, anti-HEV GMC remained high till 43 months of follow-up, irrespective of the number of doses received, whether one, two, or three. These data suggest that two doses of the HEV 239 vaccine in a 0–1-month schedule are highly likely to provide durable anti-HEV antibody concentration and, hence, protection against HEV infection and disease in HEV-seropositive persons.

By contrast, in individuals who were seronegative for anti-HEV at baseline, the anti-HEV levels reached immediately after vaccination and at all time points thereafter till 43 months were lower after two vaccine doses than after the usual three doses. However, notably, the GMC following the two-dose schedule in the participants who were seronegative at baseline was quite high till at least 13 months of follow-up, being higher at this time point than the GMC observed at 43 months following the usual three-dose schedule. This duration of antibody persistence may be quite sufficient for controlling a disease outbreak, as further discussed below.

In a trial in Bangladesh, which specifically assessed a short two-dose schedule (0–1 months) of HEV 239 [14], seropositivity rates and antibody titers were fairly high, i.e., consistent with the data from the phase III trial. Unfortunately, this study lacked a group with the three-dose schedule, and hence, a head-to-head comparison of the two-dose vs. three-dose schedule of HEV 239 was not available.

Another alternative accelerated schedule with three doses given at 0–7–21 days has also been assessed in one study [13] through comparison with the standard three-dose (0–1–6 month) schedule. This accelerated schedule also led to 100% seropositivity, with antibody GMC only marginally lower than the standard regimen, suggesting that this schedule can be an option in situations necessitating the use of a shorter regimen. However, this schedule, though as quick as the two-dose (0–1 month) schedule, would be costlier and programmatically more challenging. Hence, the two-dose regimen (0–1 month) would appear to score over the 21-day three-dose schedule.

Beyond the immunogenicity data, data on efficacy are also available from three studies [6,9,14]. With regards to protection against sub-clinical infection, the two-dose schedule of HEV 239 in phase II pre-licensure study [9] and the Bangladesh trial [14] showed efficacy rates of 85% and 100%, respectively. However, the interpretation of these studies is limited by the fact that they did not look at clinical disease and instead looked at serological evidence of HEV infection. By comparison, in the phase III licensure trial, efficacy was assessed as protection against clinical disease. In this trial, the protection following two doses was found to be 100%, albeit with wide confidence intervals, and this lasted till at least 30 months after vaccination [6]; after this time point, the low number of cases precluded any reliable conclusion. Even after 30 months, a comparison with a post-hoc general population control group showed that the two-dose schedule was a good protection; however, for this post-hoc comparison, a possibility of bias cannot be excluded. Even if one were to ignore this delayed protection, the 30-month duration of protection is by itself sufficient in situations where a rapid schedule is most required, such as for the control of a hepatitis E outbreak.

In fact, in parts of the world where outbreaks of hepatitis E occur, the background circulation of HEV is higher, and the baseline anti-HEV seropositivity rate is likely to be higher. Given that (i) in seropositive persons, even one dose of HEV 239 leads to a long-duration boosting of anti-HEV antibodies persisting beyond 43 months, and (ii) in seronegative persons, two doses of HEV 239 lead to fairly high anti-HEV GMC at least till 13 months, it appears quite reasonable to use two-dose (0, 1 month) schedule to control hepatitis E outbreaks in disease-endemic areas.

HEV vaccination is expected, in addition to the induction of specific antibody responses, to induce specific cellular immune responses, which are likely to play a complementary role in protection against HEV infection. However, these responses have not been studied in detail in the available studies, either following the usual three-dose schedule or fewer doses.

Further, our analysis shows that the two-dose schedule was as safe as the three-dose schedule. Though these data are based on a small-size study, a schedule with fewer doses would, in any case, not be expected to raise a safety concern.

In conclusion, our systematic review strongly supports the use of a rapid or a shortened two-dose schedule for the HEV 239 vaccine for the control of hepatitis E outbreaks.

## Figures and Tables

**Figure 1 vaccines-13-00028-f001:**
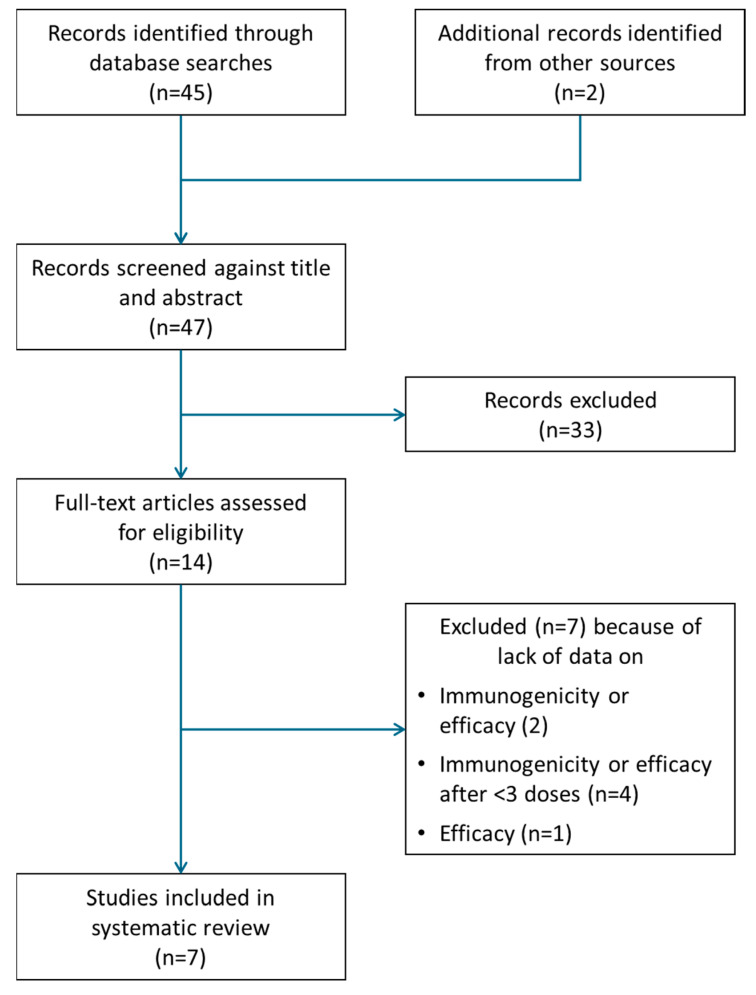
Flow chart showing the process of the literature search and screening for eligibility.

**Figure 2 vaccines-13-00028-f002:**
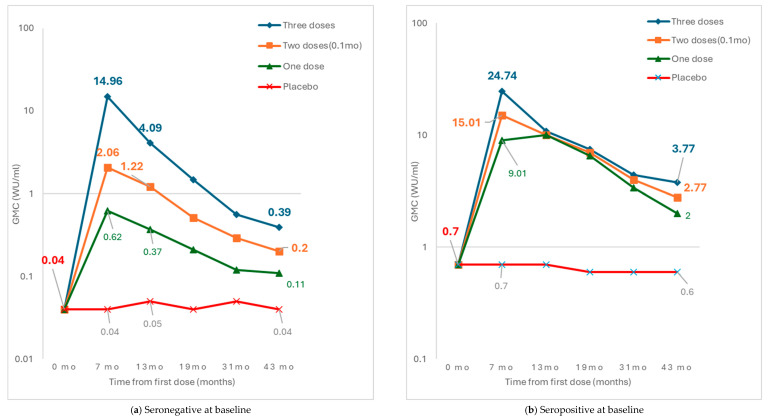
Geometric mean concentration of IgG anti-HEV and seropositivity rates among participants in immunogenicity sub-group. The left panel shows data for those who were seronegative for anti-HEV antibodies at baseline, and the right panel shows data for those who were seropositive at baseline.

**Table 1 vaccines-13-00028-t001:** Inclusion and exclusion criteria applied to the results of the literature search.

**Inclusion Criteria**
Contains data on immunogenicity, efficacy, or effectiveness of *Escherichia coli* expressed HEV 239 virus-like particle (VLP) recombinant protein vaccine (Hecolin^®^), with fewer than 3 doses or a shorter time interval between doses;
Either a clinical trial or observational study;
Populations: All;
Language restriction: None.
**Exclusion Criteria**
Not hepatitis E vaccine or Hecolin^®^;
Non-human studies;
Pre-clinical studies, systematic reviews, and the grey literature.

**Table 2 vaccines-13-00028-t002:** List of studies identified for inclusion in the review.

Author, Year (Reference Number); Location	Brief Description	Type of Data Included	Number of Planned Doses/Schedules	Characteristics of Study Subjects	Number of Subjects
**During Initial Vaccine Development**					
Zhang et al., 2009 [9]; China	Pre-licensure phase II study	Immunogenicity, efficacy	2 doses (0, 6 months) vs. 3 doses (0, 1, and 6 months)	Healthy adults (age 16–55 years); seronegative	457
Zhu et al., 2010 [11]; China Zhang et al., 2015 [12]; China Huang et al., 2024 [6]; China	Phase III (licensure) study Longer term follow-up (4.5 years) of phase III study Longer term follow-up (10 years) of phase III study	Immunogenicity, efficacy	3 doses (0, 1, and 6 months); however, it includes data on immunogenicity and efficacy in some subjects who received fewer doses	Community volunteers aged 16–65 years; irrespective of HEV serostatus (47% seropositive)	112,604
**Post-Licensure Studies**					
Chen et al., 2019 [13]; China	Clinical trial of a 3-dose accelerated schedule (0, 7, and 21 days)	Immunogenicity	3 doses in accelerated (0, 7, and 21 days) vs. usual (0, 1, and 6 months) schedule	Healthy adults aged 18 years or older; seronegative	126
Øverbø et al., 2023 [14]; Bangladesh	Clinical trial of a 2-dose schedule versus a 3-dose schedule of hepatitis B vaccine	Immunogenicity, efficacy	2 doses (0 and 1 months)	Healthy volunteers aged 16–39 years, irrespective of serostatus (33% seropositive)	100
Kao et al., 2024 [15]; USA	Phase I study for the United States population	Immunogenicity	3 doses (0, 1, and 6 month); includes data on antibody response after 1 or 2 doses.	Healthy adults in the United States; seronegative	25

**Table 3 vaccines-13-00028-t003:** Results of seroconversion and antibody titers at 7 months after starting vaccination in phase II trial of HEV 239 vaccine (Zhang et al., 2009) [9].

Vaccine Administered and the Dose Schedule	*n*	Seropositivity (%)	IgG Anti-HEV, WU/mL; Geometric Mean Concentration (95% CI)
HEV 239; 20 μg × 3 doses (0, 1, and 6 months)	128	100	15.9 (13.8–18.2)
HEV 239; 20 μg × 2 doses (0 and 6 months)	109	98	8.6 (6.5–11.3)
HBV vaccine; 5 μg × 3 doses (0, 1, and 6 months)	131	8	-

**Table 4 vaccines-13-00028-t004:** Results of seroconversion and antibody titers at 7 and 44 months after starting vaccination in the phase III trial of HEV 239 vaccine (Zhu et al., 2010 [11]; Zhang et al., 2015 [12]).

Number of Doses Received and Time Since the First Dose	Three Doses (0, 1, 6 Month)	Two Doses (0, 1 Month)	Two Doses (0, 6 Month)	One Dose (0 Month)
**Seronegative at Baseline**				
*Seropositivity rate*				
7 months	99.9%	98%	100%	98%
13 months	100%	100%	100%	100%
43 months	92%	82%	80%	39%
*Anti-HEV antibody titer (WHO Units/mL), Geometric mean concentration (95% confidence interval)*				
7 months	14.96 (14.48–15.45)	2.06 (1.68–2.51)	7.03 (5.67–8.72)	0.62 (0.45–0.87)
13 months	4.09 (3.81–4.39)	1.22 (0.61–2.43)	2.13 (1.58–2.87)	0.37 (0.13–1.11)
43 months	0.39 (0.37–0.41)	0.20 (0.14–0.29)	0.21 (0.15–0.29)	0.11 (0.05–0.24)
**Seropositive at Baseline**				
*Seropositivity rate*				
7 months	-	-	-	-
43 months	99.8%	100%	100%	100%
*Anti-HEV antibody titer (WHO Units/mL), Geometric mean concentration (95% confidence interval)*				
7 months	24.74 (14.74–34.74)	15.01 (12.01–17.01)	24.54 (14.54–34.54)	9.01 (7.01–10.01)
43 months	3.77 (2.77–4.77)	2.77 (1.77–3.77)	3.57 (2.57–4.57)	2.00 (1.34–2.77)

**Table 5 vaccines-13-00028-t005:** Results of seroconversion and antibody titers at 2 and 24 months after starting vaccination in the Bangladesh trial of HEV 239 vaccine (Øverbø et al., Vaccine 2023) [14].

Time Since First Dose	Seropositivity Rate	Antibody Titer (WHO Units/mL), Geometric Mean Concentration (95% Confidence Interval)
** *Overall (n = 50)* **		
2 months	100%	49.5 (31.5–77.7)
24 months	98%	6.4 (3.6–11.3)
** *Seropositive at baseline (n = 20)* **		
2 months	100%	189.3 (122.3–293.1)
24 months	100%	36.1 (23.2–56.4)
** *Seronegative at baseline (n = 30)* **		
2 months	100%	20.2 (12.6–32.5)
24 months	95%	1.6 (1.0–2.5)

**Table 6 vaccines-13-00028-t006:** Long-term vaccine efficacy in participants administered with two doses (first and second dose) (Huang et al., 2024) [6].

Time Period * (Months from the Start of Vaccination)	Disease Incidence (Per 100,000 Person Years)			Vaccine Efficacy, % (95% Confidence Interval) **	
	HEV 239 Vaccine Group (*n* = 3542)	Placebo Group (*n* = 3765)	Post-Hoc External Control Group (*n* = 178,236)	Vaccine vs. Placebo	Vaccine vs. External Control Cohort *
1.5–30	0.0	2.3	2.0	100 (+15.4 to 100)	100 (31.1 to 100)
1.5–54	0.3	1.8	2.1	83.4 (−36.7 to 99.6)	85.7 (19.1 to 99.6)
1.5–78	0.2	1.5	2.0	83.4 (−36.5 to 99.6)	88.1 (32.8 to 99.7)
1.5–102	0.2	1.5	1.9	85.8 (−10.4 to 99.7)	89.1 (39.0 to 99.7)
1.5–120	0.2	1.3	1.8	85.8 (−10.4 to 99.7)	89.9 (43.4 to 99.7)

* The time periods relate to the time period beginning with the first vaccine dose over which efficacy was assessed. These begin at 1.5 months, i.e., 15 days after the 2nd dose, which was administered one month after the first dose, and continue for variable time after vaccination. ** The vaccine efficacy results are from post-hoc analysis and are standardized for age and sex.

## Data Availability

Data were retrieved from the articles mentioned in the references.

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
