# Peer review of "Immunogenicity, Efficacy, and Effectiveness of Two-Dose and Shorter Schedules of Hepatitis E Vaccine: A Systematic Review"

_vaccines, 2024, doi:10.3390/vaccines13010028_

Round 1

Reviewer 1 Report

Comments and Suggestions for Authors

In this paper, Azam et al. designed a systematic review to investigate the possibility of shorter vaccination protocols for the "HEV 239" vaccine. Although the article is well-written and the study is well-designed, it is very descriptive, which does not hinder its publication. Here are a few comments below that I think could improve the article:

- The discussion session could be shortened and better structured. 

- Include a paragraph listing and discussing the limitations in the discussion. Some limitations I noticed were the lack of data from other vaccines to compare and the low incidence of HEV during the vaccine efficacy study.

- The absence of correlates of protection (briefly mentioned at the beginning of the discussion) and the lack of data on the importance of cellular immunity (not considered in the systematic review design) need to be explored further. 

- Lines 223-226: Where do the criteria to define Hepatitis E illness come from? Please include the citation.

Minor points:

Which author is "MBA"? Cited in "author contribution", "funding" and "conflict of interest"

Author Response

Comment 1: In this paper, Azam et al. designed a systematic review to investigate the possibility of shorter vaccination protocols for the "HEV 239" vaccine. Although the article is well-written and the study is well-designed, it is very descriptive, which does not hinder its publication. Here are a few comments below that I think could improve the article:

Response 1: We thank the reviewer for the above comments.

Comment 2: The discussion session could be shortened and better structured. 

Response 2: We thank the reviewer for the input. We have reviewed the ‘Discussion’ section. We find that it is only about 1.5 pages long (the entire manuscript is 14 pages), and hence feel that the length of this section is reasonable. We tried to shorten it; however, we were unable to do so without losing some useful arguments. We are open to the idea of revising this section in keeping with any specific suggestions that the reviewer may have.

Comment 3: Include a paragraph listing and discussing the limitations in the discussion. Some limitations I noticed were the lack of data from other vaccines to compare and the low incidence of HEV during the vaccine efficacy study.

Response 3: The limitations of individual studies have already been indicated in the discussion section. These appear at lines 304-309, 322-324, 340-342, 358 and 362 in the revised manuscript.

Comment 4: The absence of correlates of protection (briefly mentioned at the beginning of the discussion) and the lack of data on the importance of cellular immunity (not considered in the systematic review design) need to be explored further.

Response 4: In general, cellular immunity following HEV vaccination has received very little attention. It was discussed only in one study from Bangladesh which compared 2-dose vs 3-dose schedule of HEV 239, but there were no data on this in this study. We have now added this as a separate paragraph in the discussion section in the revised manuscript (lines 373-376).  

Comment 5: Lines 223-226: Where do the criteria to define Hepatitis E illness come from? Please include the citation.

Response 5: These are from Reference 11. This reference number has now been added to the manuscript at this location (now line 231-235 – since the text has moved down due to addition of some text in the introduction section, as per suggestions of the other reviewers).

Comment 6: Minor points: Which author is "MBA"? Cited in "author contribution", "funding" and "conflict of interest".

Response 6: We thank the reviewer for picking up this discrepancy. It has now been changed to BA (the first author).

Reviewer 2 Report

Comments and Suggestions for Authors

The review by Bilal Azam et al entitled “Immunogenicity, Efficacy and Effectiveness of Two-Dose and 2 Shorter Schedules of Hepatitis E Vaccine: A systematic review” addresses an interesting an relevant topic. At present the HEV vaccine Hecolin which is licensed in China, Pakistan and India is administered based on a three dose schedule. As stated by the authors this schedule might not be adequate for the control of acute outbreaks. In light of this the authors performed a systemtic review  of immunogenicity and effectiveness data in the context of shorter-duration vaccination schedules.
There are few points that should be addressed:
-A few more information about HEV structure and genotypes should be included
-The authors should describe the rationale for the established immunization schedule (0, 1 and 6 month) with Hecolin
-the authors should provide the reasons for the lack of approval in the EU for Hecolin and US
- describing the immunogenicity data, the authors should shortly clarify the assay systems used for the quantification of orf2-specific antibodies
-with respect to the immunogenicity data the authors should include a statement about the relevance of the antibody titer and the neutralizing capacity  of different genotypes. In light of the lack of an established correlate of protection the authors should include considerations about the relevance of antibody titer data with respect to the protective potential and with respect to the protection from different genotypes.
-in section 3.3 the different endpoints of the efficacy studies described in this section should be presented more clearly. Moreover, the genotype of the HEV causing illness in the cited studies should be mentioned.
-

Author Response

Comment 1: The review by Bilal Azam et al entitled “Immunogenicity, Efficacy and Effectiveness of Two-Dose and 2 Shorter Schedules of Hepatitis E Vaccine: A systematic review” addresses an interesting an relevant topic. At present the HEV vaccine Hecolin which is licensed in China, Pakistan and India is administered based on a three dose schedule. As stated by the authors this schedule might not be adequate for the control of acute outbreaks. In light of this the authors performed a systematic review of immunogenicity and effectiveness data in the context of shorter-duration vaccination schedules.

Response 1: No response needed.  

Comment 2: There are few points that should be addressed:
A few more information about HEV structure and genotypes should be included

Response 2: This has been added to the introduction section (lines 40-46).
This has also led to a change in reference 2.

Comment 3: The authors should describe the rationale for the established immunization schedule (0, 1 and 6 month) with Hecolin.

Response 3: Choice of the three-dose schedule for HEV 239 was based on a phase II study in China [Zhang et al, 2009 (10)], which showed higher antibody titers after three  doses than with two doses. We have now added this to the ‘Introduction’ section of our manuscript, as per the reviewer’s suggestion (lines 57-61).

This has led to a change in the sequence of references (references 9 and 10).

Comment 4: The authors should provide the reasons for the lack of approval in the EU for Hecolin and US.

Response 4: To the best of our knowledge, the manufacturer has not yet submitted the product for approval in these countries.

Comment 5: Describing the immunogenicity data, the authors should shortly clarify the assay systems used for the quantification of orf2-specific antibodies.

Response 5: As already indicated in the text of our manuscript, an enzyme immunoassay from Wantai, China was used.

Comment 6: With respect to the immunogenicity data the authors should include a statement about the relevance of the antibody titer and the neutralizing capacity of different genotypes. In light of the lack of an established correlate of protection the authors should include considerations about the relevance of antibody titer data with respect to the protective potential and with respect to the protection from different genotypes.

Response 6: This is already included in the discussion section of the manuscript (lines 298-301). This has also led to addition of two references [17,18].

Comment 7: In section 3.3 the different endpoints of the efficacy studies described in this section should be presented more clearly. Moreover, the genotype of the HEV causing illness in the cited studies should be mentioned.

Response 7: Most of the studies do not provide information on the HEV genotype causing infection among vaccinees or controls. The information is available for one study and this has now been added (please see lines 267-270 of the revised manuscript). Also, consequently (as also referred to above), text has been added to the discussion too (lines 298-301, and references 17 and 18).

Reviewer 3 Report

Comments and Suggestions for Authors

This study reviews the immunogenicity and efficacy of Hepatitis E vaccines in three or two administration schedules.

The review is very thorough and arrives to interesting and useful conclusions.

The manuscript is well organized and makes easy to understand the different aspects to consider in order to draw conclusions.

Minor points to point out:

1-      Table 2, it could be useful to include the number of patients of each study, even though this aspect is included in the text

2-      Do any of the selected articles give information about adverse effects? Or differences in adverse effects between the different schedules?

Author Response

Comment 1: This study reviews the immunogenicity and efficacy of Hepatitis E vaccines in three or two administration schedules.

The review is very thorough and arrives to interesting and useful conclusions.

The manuscript is well organized and makes easy to understand the different aspects to consider in order to draw conclusions.

Response 1: We thank the reviewer for the positive comments. No response is needed.  

Comment 2: Minor points to point out:

Table 2, it could be useful to include the number of patients of each study, even though this aspect is included in the text.

Response 2: The number of subjects in each study had already been included in the last column of Table 2.

Comment 3: Do any of the selected articles give information about adverse effects? Or differences in adverse effects between the different schedules?

Response 3: In a pre-licensure study [Zhang et al, 2009 (10)] in China that undertook a head-to-head comparison of 2-dose (at 0 and 6 months) vs 3-dose (at 0,1 and 6 months) schedules of HEV 239 vaccine (at a dose of 20 µg instead of the 30 µg dose in the current product), no significant difference was reported in the frequencies of grade 3 local or systemic reactions between these dose-schedules. In Chen et al, 2019 (13), an accelerated 3-dose schedule (0, 7 and 21 days) was compared with the routine 3-dose schedule (0, 1 and 6 months) of HEV 239 vaccine, no significant difference was found in the incidence rate of solicited adverse events between the groups. No other study reports a comparison of adverse events between 3-dose and 2-dose schedules.

We believe that a vaccine schedule with fewer doses would not generally be expected to lead to increased adverse events. This is the reason that we have not included the safety data in our manuscript.

Round 2

Reviewer 2 Report

Comments and Suggestions for Authors

the authors adressed all points

Author Response

Comment: the authors addressed all points

Response: No reply is needed.